# Biocompatibility Parameters with Standard and Increased Dose of Citrate in Hemodialysis—A Randomized Trial

**DOI:** 10.3390/jcm10132987

**Published:** 2021-07-04

**Authors:** Alesa Orsag, Mojca Bozic-Mijovski, Samo Hudoklin, Sasa Simcic, Jakob Gubensek

**Affiliations:** 1Department of Nephrology, University Medical Center Ljubljana, 1000 Ljubljana, Slovenia; alesaorsag@gmail.com; 2Faculty of Medicine, University of Ljubljana, 1000 Ljubljana, Slovenia; 3Department of Vascular Diseases, University Medical Center Ljubljana, 1000 Ljubljana, Slovenia; mojca.bozic@kclj.si; 4Faculty of Medicine, Institute of Cell Biology, University of Ljubljana, 1000 Ljubljana, Slovenia; samo.hudoklin@mf.uni-lj.si; 5Faculty of Medicine, Institute of Microbiology and Immunology, University of Ljubljana, 1000 Ljubljana, Slovenia; Sasa.Simcic@mf.uni-lj.si

**Keywords:** biocompatibility, complement system, hemodialysis, myeloperoxidase, platelet factor 4, regional citrate anticoagulation, thrombin–antithrombin complex

## Abstract

Background: The dose of citrate needed in regional citrate anticoagulation (RCA) to achieve optimal biocompatibility is unknown. We performed a randomized trial comparing two doses (ACTRN12613001340729). Methods: In 30 patients a single hemodialysis with either standard (2.7 mmol/L) or increased dose of citrate (4 mmol/L) was performed. C5a-desArg, myeloperoxidase (MPO), thrombin-antithrombin complex (TAT), and platelet factor 4 (PF4) were measured and the inner surface of the dialyzer fibers was evaluated with scanning electron microscopy (SEM). Results: A good separation of anticoagulation effect was achieved (post-filter ionized calcium 0.20 vs. 0.31 mmol/L, *p* < 0.05). There was no effect of citrate dose on any of the biocompatibility parameters; transient and parallel increase in PF4 after 30 min and parallel increase in TAT after 4 h were observed. There were no visually detected clotting problems within the circuit and no significant hypocalcemia in either group. SEM clotting score was excellent and comparable in both groups (*p* = 0.59). Conclusions: Given the excellent results in both groups, absence of between group differences and inability of the increased dose of citrate to completely blunt the small residual increase in PF4 and TAT, we conclude that the standard dose of citrate seems sufficient in RCA for chronic hemodialysis.

## 1. Introduction

Biocompatibility of a hemodialysis procedure refers to an (undesired) activation of several pathways in the patient as a result of the interaction of the patient’s blood with air and artificial surfaces, as well as other factors related to the hemodialysis procedure (e.g., shear stress, formation of microbubbles, etc.). Shortly after the first contact of blood with an artificial membrane, activation of complement and coagulation system occurs [1,2], leading also to activation of blood cells. Furthermore, there is a cross-activation of hemostasis and inflammation [3,4]. Not only does this result in suboptimal anticoagulation of the extracorporeal circuit, affecting the efficiency of the dialysis procedure, but uncontrolled activation of complement and hemostasis has also been associated with chronic microinflammation, increased oxidative stress and thrombotic diathesis. By these mechanisms persistent exposure to bio-incompatible hemodialysis can lead to accelerated cardiovascular disease [1,5,6] and even increased mortality in dialysis patients [1,2]. Therefore, improved biocompatibility of the hemodialysis procedure is of great importance for the long-term survival of chronic hemodialysis patients.

Biocompatibility of hemodialysis can be improved by modern membrane materials and dialysis circuit configuration, but also by the choice of anticoagulation during dialysis, which affects the residual bio-incompatibility of the materials used. Regional citrate anticoagulation (RCA) has become a widely used method of anticoagulation in extracorporeal blood purification [7,8]. The anticoagulant effect in RCA is mediated by severe hypocalcemia in the extracorporeal circuit, which also prevents cellular activation and improves biocompatibility. More specifically, RCA has been shown to reduce platelet and leukocyte activation [9,10,11], as well as complement activation [12].

Many different protocols have been used in clinical practice for RCA and the “standard” (i.e., most often used) dose of citrate in the extracorporeal circuit is about 2.4–3 mmol citrate/L of blood [13,14,15], but higher doses (3.4–4.2 mmol/L [9,11,16]) have also been used. In vitro evidence shows that many inhibitory effects of citrate in the extracorporeal circuit are dose-dependent [17] and are a result of the level of ionized hypocalcemia achieved in the circuit [18], i.e., more severe ionized hypocalcemia more intensely inhibits the blood cells’ interactions with the dialysis membrane. Clinical data also suggest that increased doses of citrate are required to abolish complement and blood cells activation [9,11], but there are no studies directly comparing different doses of citrate in a clinical setting. Therefore, the optimal dosage of citrate in chronic hemodialysis with regard to biocompatibility remains to be determined.

The aim of our study was to compare a standard and an increased citrate dose in RCA in a clinical setting with respect to biocompatibility parameters.

## 2. Patients and Methods

### 2.1. Study Design and Patients

We conducted a randomized controlled trial comparing a standard (2.7 mmol/L) and an increased dose (4 mmol/L) of citrate in RCA, enrolling 30 adult chronic hemodialysis patients. We excluded patients with clinical signs of infection, acute bleeding, malignancy, as well as the use of immunosuppressive, antithrombotic or anticoagulant drugs, as these could increase some of the measured biocompatibility parameters. After obtaining a written informed consent, each patient was randomly assigned to a single hemodialysis session of 4 h duration with either a standard dose (2.7 mmol/L) or an increased dose (4 mmol/L) of citrate. Randomization was conducted using a simple randomization list (created from http://www.randomization.com/ (accessed on 1 May 2018) and there was no allocation concealment. The study was performed from January to June 2018 at the University Medical Center, Ljubljana. The study was approved by the National Medical Ethics Committee (Ref. No. 89/03/13) of the Republic of Slovenia and registered at Australian New Zealand Clinical Trials Registry (ACTRN12613001340729, date of registration 9 December 2013). All experiments were performed in accordance with relevant guidelines and regulations. Sample size was calculated based on available data on thrombin–antithrombin complex (TAT) during RCA, which reached 22 ng/mL by the end of the dialysis procedure at standard dose of citrate [14]. Assuming 50% lower values of TAT in the increased dose group and 80% power, a sample size of 26 was obtained, which was rounded up to 30 procedures.

### 2.2. Dialysis Procedure and RCA

Hemodialysis was performed with synthetic high-flux dialyzers FX Cordiax (Fresenius Medical Care, Bad Homburg, Germany) or Polyflux H (Baxter Healthcare Corporation, Deerfield, MA, USA) as prescribed for each patient’s regular hemodialysis. Blood flow was maintained at 250 mL/min throughout the dialysis procedure. RCA was achieved with an 8% trisodium citrate solution infused continuously into the arterial line at a flow rate of 150 mL/min in the standard dose group and 220 mL/min in the increased citrate dose group, resulting in a calculated citrate concentration of 2.7 mmol/L of blood and 4 mmol/L, respectively (see Table 1). We used a calcium-free dialysate at a flow rate of 500 mL/min. Calcium was substituted using 1 M calcium chloride infused into the venous line, starting at 14 mL/min in the standard dose and 16 mL/min in the increased dose group and adjusted to maintain ionized calcium (iCa) in the desired target range (1.0–1.2 mmol/L).

### 2.3. Standard Laboratory and Biocompatibility Parameters

Regarding standard laboratory parameters, magnesium, total and corrected calcium were measured before and after dialysis in our hospital’s laboratory. Blood gas analysis was performed before and after dialysis and iCa was measured hourly on a point-of-care ABL ABL800 Flex analyzer (Radiometer Medical, Bronshoj, Denmark).

For biocompatibility parameters, blood was drawn from the vascular access (AVF or hemodialysis catheter) prior to initiation of hemodialysis (t_0_), after 30 min from the venous line (t_30V_) and after 4 h of hemodialysis from the arterial (t_4A_) and venous lines (t_4V_) of the dialysis system. In patients with hemodialysis catheters, 20 mL of blood were aspirated from the catheter before taking the t_0_ sample to exclude the possibility of contamination with the locking solution (30% sodium citrate). During and after hemodialysis, we briefly interrupted the infusion of 8% trisodium citrate and calcium before taking the sample. All samples were transferred to the laboratory within one hour (on ice, when necessary), centrifuged, aliquoted and stored at −80 °C until analysis. Hematocrit (Ht) was also measured at all time points. To eliminate the effect of hemoconcentration as a result of ultrafiltration during hemodialysis, all biocompatibility parameters measured after the start of dialysis were corrected to baseline Ht (t_0_) using the following formula: c_1′_ = c_1_ * (Ht_0_ (1−Ht_1_)/Ht_1_ (1−Ht_0_)) [19].

The primary outcomes of the study were biocompatibility parameters. Activation of the coagulation cascade was assessed by measuring TAT and platelet activation by serum platelet factor 4 (PF4) concentration. Samples for both parameters were collected into 4.5 mL tubes containing citrate-theophylline-adenine-dipyridamole (CTAD, Becton Dickinsen, Franklin Lakes, New Jersey, USA) and placed in ice-cold water immediately after sampling to prevent activation of platelets and coagulation. Plasma was prepared by a 20-min centrifugation at 2000× *g*, snap frozen in liquid nitrogen and stored at ≤−70 °C until analysis. For analysis we used an enzyme immunoassay (ASSERACHROM^®^ PF4, Diagnostica Stago, Asnieres-sur-Seine, France, and Enzygnost^®^ TAT micro, Siemens Healthcare Diagnostics Products GmbH, Marburg, Germany). Leukocyte activation, more specifically the activation of neutrophils and the release of granule enzymes, was evaluated by measuring serum myeloperoxidase (MPO) with an enzyme-linked immunosorbent assay (ELISA), according to the manufacturer’s instructions (Immundiagnostik AG, Bensheim, Germany). The detection limit of this assay was 30 ng/mL. Complement activation was assessed by measurement of C5a-desArg in blood plasma with BD OptEIA^TM^ Human C5a ELISA, according to the manufacturer’s instructions (Becton, Dickinson and Company BD Biosciences, San Jose, CA, USA). Blood samples were collected into tubes containing ethylenediamine tetra-acetic acid (EDTA) and placed in ice-cold water to prevent complement activation after sampling. Samples were centrifuged at 4 °C for 15 min at 1000× *g* and plasma samples were stored at −80 °C until testing.

### 2.4. Scanning Electron Microscopy

As a secondary outcome, adhesion of blood cells and activation of clotting on inner surfaces of dialyze fibers were visualized using scanning electron microscopy (SEM). Five dialyzers were randomly selected from each group and rinsed with 3 L of isotonic saline immediately after hemodialysis procedure. Each dialyzer was cut with a saw and some of the fibers were transferred to 3% formaldehyde and 2% glutaraldehyde in 0.1 M cacodylate buffer (pH = 7.4) for 2 h 45 min. Fibers were washed in 0.1 M cacodylate buffer and postfixed in 1% osmium tetroxide in the same buffer for 1 h at 4 °C. Fibers were then dehydrated in an ethanol series and dried with hexamethyl-disilazane. A 3–5 mm long section of 10 randomly selected fibers per patient was cut open lengthwise with a scalpel under a stereomicroscope to expose the inner surface of the dialysis membrane (which was in contact with blood) and was attached on an aluminum holder with carbon adhesive tabs. Samples were then sputter-coated with gold and examined with a Vega3 scanning electron microscope (Tescan, Brno-Kohoutovice, Czech Republic) running at 25 kV.

For evaluation of SEM images, we used a slightly modified, previously used semi quantitative scoring system [17,20], where we evaluated: (a) fiber area covered by adherent material/cells, (b) fibrin network deposition, (c) erythrocyte aggregates, (d) platelet adhesion, and (e) obstruction of fiber lumen. All five parameters were quantified on a semiquantitative scale of 0–4 corresponding to the estimated covered surface area: 0-no signs of adhesion, 1–<25% of visible surface, 2–25%–50%, 3–50%–75% and 4–>75% of visible surface. Individual scores were summed to obtain a SEM clotting score (range 0–20).

### 2.5. Statistical Analysis

The results are given as mean ± standard deviation (SD) or as median and interquartile range (IQR), as appropriate. Variables were compared between the groups using Student’s *t* test. Biocompatibility parameters were analyzed using repeated measures ANOVA, with different time points as a repeated measures factor, dose group as a categorical factor and an interaction effect. A Bonferroni post-hoc test was used to identify significant differences between different levels of the repeated measures factor. SEM clotting score was compared between groups with Mann-Whiney U-Test. All analyzes were performed with Statistica 7.0 software (StatSoft, Inc., Tulsa, OK, USA). A value of *p* < 0.05 was considered statistically significant.

## 3. Results

### 3.1. Patient Characteristics

We enrolled 30 patients, 15 patients in each group (see Figure 1). There were nine men and six women in the standard citrate dose group, with a mean age of 62 ± 14 years and a mean renal replacement therapy vintage of 6 ± 2.5 years. In the increased citrate dose group there were 10 men and five women, with a mean age of 71 ± 13 and a mean renal replacement therapy vintage of 11 ± 10 years (see Table 2).

### 3.2. Basic Laboratory Parameters, Anticoagulation Efficiency and Complications

Basic laboratory parameters are given in Table 3 and were comparable between the groups. Different citrate doses in both groups led to lower post-filter ionized calcium (iCa) values in the increased dose group: 0.23 mmol/L vs. 0.32 mmol/L after 30 min (*p* < 0.05) and 0.20 mmol/L vs. 0.30 mmol/L after 2 h (*p* < 0.05). There were no clotting problems during hemodialysis procedures and all dialysis systems were visually clean after dialysis.

Regarding complications, there were no cases of significant hypocalcemia (iCa < 0.9 mmol/L) during dialysis procedures, although one patient started dialysis with an iCa of 0.82 mmol/L in the standard citrate dose group, which then increased during dialysis. There were five cases of significant metabolic alkalosis (pH > 7.5)–4 in the standard group and one case in the increased dose group. Total-to-ionized calcium ratio remained normal in all patients indicating that no citrate accumulation occurred.

### 3.3. Biocompatibility Parameters

There was a mild, but statistically significant, decrease in leukocytes from 30 min onward (see Figure 2A); the decrease was comparable in both groups (significant effect of time (*p* < 0.01), but no effect of dose group). Thrombocyte values in samples from arterial line were stable, but there was a mild increase in thrombocytes from arterial to venous sampling site at 4 h of dialysis (see Figure 2B); the increase was comparable in both groups (significant effect of time (*p* < 0.01), but no effect of dose group). There was no significant increase of C5a-desArg level during dialysis (see Figure 2C) and no between group differences (no effect of time or dose group). We observed a significant increase in PF4 after 30 min of hemodialysis, which decreased to pre-dialysis levels by the end of hemodialysis (see Figure 2D); the increase was comparable in both groups (significant effect of time (*p* < 0.01), but no effect of dose group). TAT levels increased after 4 h in both groups, but only in the arterial line samples (see Figure 2E). The increase was comparable in both groups (significant effect of time (*p* < 0.01), but no effect of dose group). There was a statistically significant but small decrease in MPO levels after 4 h of dialysis in both groups, but only in the arterial line samples (see Figure 2F). The decrease was comparable in both groups (significant effect of time (*p* < 0.01), but no effect of dose group).

### 3.4. Electron Microscopy of Dialyzer Fibers

SEM proved to be a suitable method to analyze the extent and nature of deposits on the dialyzer fibers. SEM revealed very low levels of cell and fibrin adhesion in both high and low citrate groups. Of all evaluated fiber areas, 25% were completely clean, with no biological deposits visible on SEM (see Figure 3A). In other cases, the most common finding, regardless of the citrate concentration, was some adherent platelets, exhibiting cytoplasmic extensions and often forming non-confluent islands and/or minor deposition of fibrin (see Figure 3B). Quantitation of the depositions by the total clotting score was 2 (IQR0–3) in the standard dose group and 2 (IQR 2–3; *p* = 0.59) in the increased citrate dose group.

## 4. Discussion

In this randomized controlled trial, we investigated whether an increased dose of citrate (4 mmol/L) would further increase biocompatibility of RCA compared to the standard dose (2.7 mmol/L of blood). While the increased dose achieved lower post-filter ionized calcium and was well tolerated by patients from a metabolic point of view, we could not demonstrate any further improvement in any of the measured biocompatibility parameters. Already the lower, standard dose of citrate completely eliminated activation of complement and leukocytes, inducing only a clinically insignificant decrease in leukocytes and transient and low level activation of platelets and coagulation cascade.

Complement activation, occurring within minutes after first blood-membrane contact, was the classic bio-incompatibility reaction that occurred during hemodialysis with cellulose membranes, causing leukopenia and dyspnea. Complement is activated by artificial surfaces via lectin and alternative pathways. The most important consequences of complement activation are induction of inflammation, promotion of coagulation and impairment of host defense due to accelerated consumption of complement proteins. Since inflammation and coagulation are involved in the pathogenesis of cardiovascular diseases, complement activation has been associated with susceptibility to cardiovascular disease in dialysis patients [2]. Over the past decades, synthetic membranes have been developed with improved biocompatibility and nowadays the activation of complement system during dialysis has only been reported to a minor degree [13], or not at all [11]. Mode of anticoagulation was shown to be important, although mainly with cellulose membranes. Reduced complement activation with citrate anticoagulation was shown in apheresis [12] and during dialysis with cellulose membranes [21], but not with modern synthetic membranes [11]. Although our previous in-vitro data showed a trend towards lower complement activation at higher doses of citrate [17], the results of the present clinical study confirm that no significant activation of complement occurs during RCA even at the standard dose.

Thrombogenicity is probably the most important characteristic of biocompatibility of an artificial material. Activation of platelets is one of the first steps in activation of hemostasis [22] and citrate was shown to completely eliminate it as compared to heparin [9]. It is known that the effect of citrate is dose-dependent. A significant increase in plasma concentration of PF4 was reported during dialysis with 2.4 mmol/L citrate [14], while there was no significant increase in PF4 at 3.4 mmol/L [9] and no increase in beta-thrombo-globulin at 4.7 mmol/L [23]. Additionally, in an in-vitro dialysis setup a significant decrease in platelet-derived micro-vesicles was observed at 4.0 mmol/L citrate compared to 3.0 mmol/L [18], while no significant increase was observed in continuous in vivo dialysis at approx. 3.0 mmol/L [24]. Contrary to our expectations, even with the increased citrate dose, we still observed a transient and small but statistically significant increase in PF4, although the absolute values were several times lower than those reported with lower citrate doses [14]. There was no decrease in thrombocytes observed during dialysis, but a mild increase in thrombocytes from arterial to venous sampling site at 4 h of dialysis, which is likely a chance finding. It therefore appears that approx. 3 mmol/L citrate is required to effectively, although not completely, abolish platelet activation. In addition to the properties of the membrane and the type of anticoagulation, there are other factors that influence platelet activation, such as blood turbulence and formation of microbubbles. Furthermore, results of TAT, an early marker of activation of the coagulation cascade, were similar: there was a parallel increase in both citrate dose groups. Data from the literature only report significant increase at the standard dose [13,14], but prior to our study, none of the studies reported TAT values at a higher citrate dose.

To comprehensively and visually evaluate platelet adhesion as well as activation of coagulation and deposition of fibrin, electron microscopy of the inner surface of dialyzer fibers has been used effectively [17,20]. A significant reduction in clotting was reported with citrate compared to low molecular weight heparins and moderate doses of unfractionated heparin [20]. Our previous in vitro results also showed a significantly better score with an increased (4 mmol/L) dose of citrate, although the results were already very good at the lower dose [17]. The results of the present randomized study could not confirm the in vitro findings and showed no further improvement of the SEM clotting score with the increased dose of citrate. The SEM clotting score was excellent and showed almost completely clean fibers in both citrate dose groups.

In addition to the activation of complement, platelets and coagulation, leukocytes are also activated during hemodialysis. The complement, hemostasis and inflammation pathways are closely intertwined and induce cross activation [2,3], which is one of the reasons for possible negative long-term effects [1,4,6]. From this aspect, it has also been reported that citrate anticoagulation is superior to heparin and attenuates leukocyte degranulation [10,11]. Citrate dose-response cannot be predicted from the available literature, as only higher doses of citrate were used in studies reporting leukocyte activation. We have observed a mild, likely clinically insignificant decrease in leukocytes observed in both groups throughout hemodialysis session, which is a well-known phenomenon. Our previous in vitro work showed that leukocyte activation was present to a comparable degree in the standard and in the increased dose of citrate [17], whereas the present in vivo study did not reveal a significant increase in MPO in either citrate group, despite mild reduction in leukocytes, providing evidence that the standard dose of citrate is sufficient in this respect.

The strengths of our study are a randomized controlled design, a good separation of the groups with respect to the anticoagulation effect (i.e., postfilter ionized calcium) and a comprehensive analysis of the biocompatibility parameters. The main limitation is a modest number of included patients and the use of different synthetic membranes.

To conclude, considering excellent results in both groups, the absence of differences between the groups and the inability of the increased dose of citrate to completely blunt the small residual increase of PF4 and TAT, the standard dose of citrate (2.7 mmol/L of blood) in RCA seems sufficient for chronic hemodialysis. Whether an even higher dose of citrate could completely blunt the remaining, small activation of hemostasis observed in this study remains to be determined. It should be noted that using a very high dose of citrate might increase the possibility of side effects and citrate accumulation, although hemodialysis provides excellent clearance of citrate. From a clinical point of view, eliminating the observed low-level residual activation of hemostasis is probably not an important treatment goal.

## Figures and Tables

**Figure 1 jcm-10-02987-f001:**
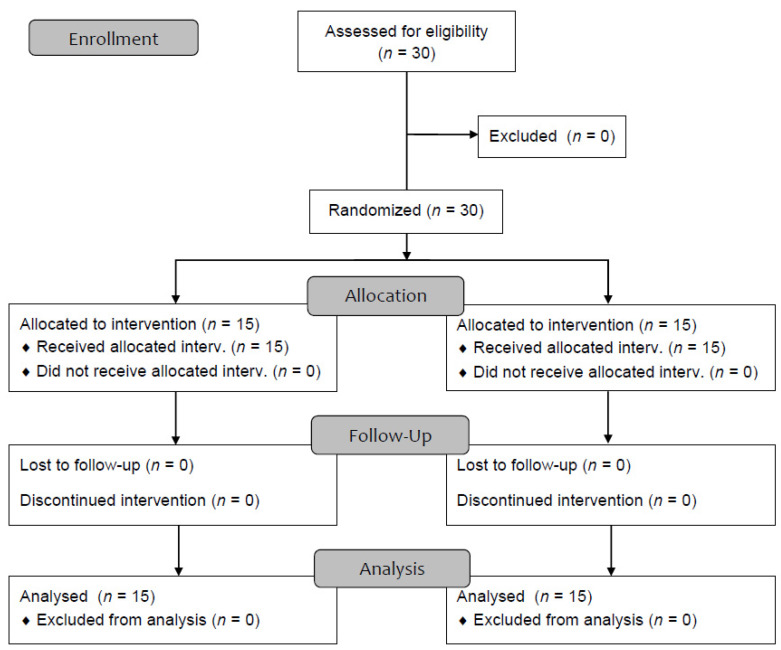
A CONSORT randomized trial flow chart.

**Figure 2 jcm-10-02987-f002:**
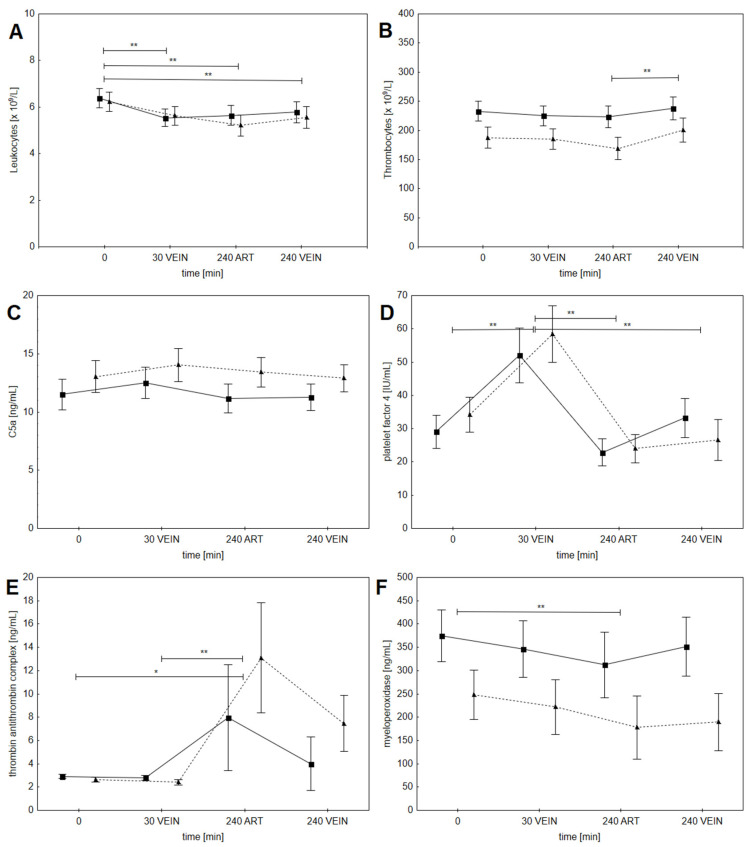
Results of repeated measures ANOVA for different biocompatibility parameters: (**A**) Leukocytes, (**B**) Thrombocytes, (**C**) complement fraction C5a, (**D**) platelet factor 4, (**E**) thrombin-antithrombin complex and (**F**) myeloperoxidase. The values shown are least-square estimates with standard error. The standard dose group is shown with squares and full line and the increased dose group with triangles and dashed line. When there was a significant effect of time, different time points were compared with a Bonferroni post-hoc test (* *p* < 0.05 and ** *p* < 0.01). There was no effect of dose group or interaction effect in any of the measured parameters.

**Figure 3 jcm-10-02987-f003:**
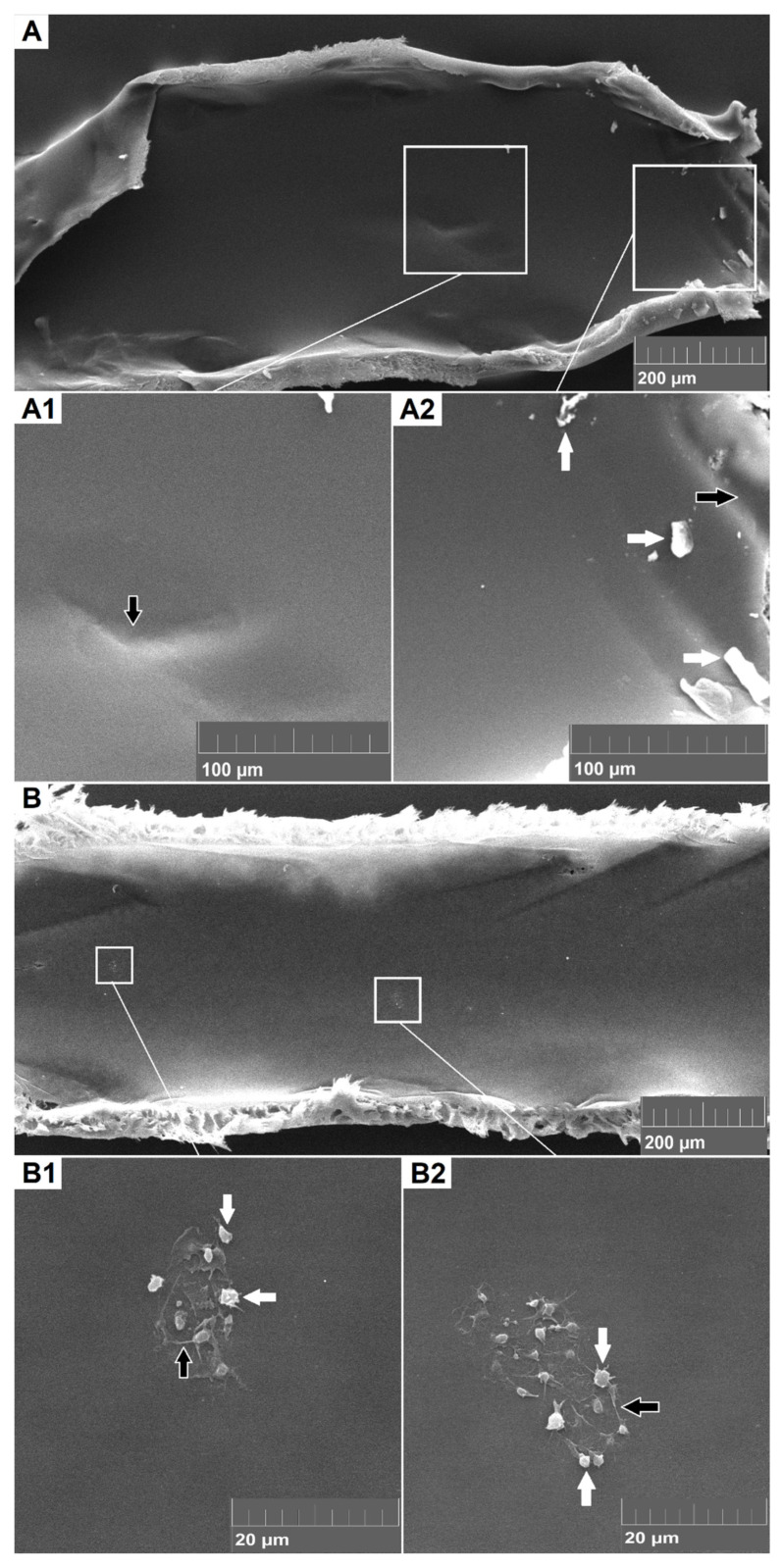
Scanning electron microscopy of the inner surface of dialyzer fibers with magnified details. (**A**) Dialyzer’s inner surface clear of biologic depositions (SEM clotting score of 0), with signs of mechanical damage, most likely caused by tweezers during fiber manipulation (black arrow in (**A1**,**A2**), and inorganic particles (i.e., debris of fiber wall), most likely produced during cutting (white arrow in (**A2**)). (**B**) Minimal depositions of platelets (white arrows in (**B1**,**B2**)) and fibrin ((black arrows in (**B1**,**B2**)) SEM clotting score of 3).

**Table 1 jcm-10-02987-t001:** Initial regional citrate anticoagulation parameters in both groups.

	Standard Dose Group	Increased Dose Group
Blood flow [mL/min]	250	250
8% Na citrate flow [mL/min]	150	220
Citrate concentration in blood * [mmol/L]	2.7	4.0
Dose of citrate [mmol/h]	40	60
Infusion of 1M CaCl_2_ ** [mL/h]	14	16

* calculated value, ** adjusted according to patient’s ionized calcium values.

**Table 2 jcm-10-02987-t002:** Baseline clinical characteristics and biocompatibility parameters in both groups. Data are presented as frequency (percentage) or mean ± standard deviation.

	Standard Dose Group	Increased Dose Group	*p* Value
N	15	15	/
Female	6 (40%)	5 (33%)	*p* = 0.75
Age	62 ± 14 years	71 ± 13 years	*p* = 0.07
Renal replacement therapy	6 ± 2.5 years	11 ± 10 years	*p* = 0.04
PF4 before hemodialysis	28 ± 16	31 ± 17	*p* = 0.33
TAT before hemodialysis (ug/L)	4 ± 4	3 ± 2	*p* = 0.28
MPO before hemodialysis (ng/mL)	342 ± 260	239 ± 70	*p* = 0.09
C5a-desArg before hemodialysis (ng/mL)	11 ± 3	11 ± 6	*p* = 0.43

**Table 3 jcm-10-02987-t003:** Comparison of basic biochemical parameters in both groups.

	Standard Dose Group	Increased Dose Group	*p* Value
corrected Ca before [mmol/L]	2.22 ± 0.23	2.21 ± 0.14	*p* = 0.45
corrected Ca after [mmol/L]	2.08 ± 0.13	2.25 ± 0.17	*p* < 0.01
total/ionized calcium ratio	2.06 ± 0.11	2.07 ± 0.11	*p* = 0.42
N of patients with total/ionized calcium > 2.5	0/15	0/15	/
post-filter ionized calcium after 30 min [mmol/L]	0.32 ± 0.05	0.23 ± 0.05	*p* < 0.01
post-filter ionized calcium after 2 h [mmol/L]	0.30 ± 0.04	0.20 ± 0.05	*p* < 0.01
Mg before [mmol/L]	1.02 ± 0.15	1.04 ± 0.14	*p* = 0.33
Mg after [mmol/L]	0.81 ± 0.06	0.80 ± 0.10	*p* = 0.47
pH before	7.44 ± 0.05	7.41± 0.04	*p* = 0.05
pH after	7.49 ± 0.05	7.46 ± 0.04	*p* = 0.06

## Data Availability

The datasets generated and analyzed during the current study are available from the corresponding author on reasonable request.

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
