# Peer review of "Biocompatibility Parameters with Standard and Increased Dose of Citrate in Hemodialysis—A Randomized Trial"

_jcm, 2021, doi:10.3390/jcm10132987_

Round 1
Reviewer 1 Report
The authors evaluated “Biocompatibility parameters with standard vs increased dose of citrate in HD”. The study has comprehensive data and is very interesting issue as a RCT, but there are some criticisms.
Major comments
- The sample size is an important limitation. Although the author calculated sample size using 22ng/mL of TAT, 50% difference between groups, and 80% power, the sample size is very small. The author may present detailed equation for sample size calculation. In fact, baseline TAT value of their study were approximately 2~3 ng/mL. Why did TAT of previous study differ from those of their study? In addition, there was a significant difference in RRT duration. Although statistical significances were not obtained, age and MPO level were marginally different between the two groups. Besides, two different membrane would be associated with biocompatibility. The author may performed multivariate analysis or subgroup analysis for overcome these limitations, but it is difficult to perform these analyses due to very small sample size.
- The reviewer understood that Figure 2 is a key result, but it is an insufficient explanation. Figure 2C shows that TAT value at 240 min was greater in increased dose group than in standard dose group. Non-significance between two groups may be caused by small sample size. In addition, the trend of MPO level was similar between two groups, but baseline MPO levels between two groups are observed. The author may point out these results.
Minor comments
- Page 2, line 29. TAT may cite the abbreviation.
- Table 2. Although the author presented accurate biocompatible markers, classical indicators, such as hs-CRP or IL-6s also need to be present.
- Type of vascular access can influence biocompatibility.
- Page 7, line 4. The author presented that 25% of all showed clean membranes. The reviewer understood that 5 membranes from each group (total 10 membrane) were evaluated in method section.
- Figure 3. Further explanations for indicators are needed.
Author Response
Point by point reply to Reviewers - Reviewer 1:
"Major comments:
The sample size is an important limitation. Although the author calculated sample size using 22ng/mL of TAT, 50% difference between groups, and 80% power, the sample size is very small. The author may present detailed equation for sample size calculation. In fact, baseline TAT value of their study were approximately 2~3 ng/mL. Why did TAT of previous study differ from those of their study? In addition, there was a significant difference in RRT duration. Although statistical significances were not obtained, age and MPO level were marginally different between the two groups. Besides, two different membrane would be associated with biocompatibility. The author may perform multivariate analysis or subgroup analysis to overcome these limitations, but it is difficult to perform these analyses due to very small sample size."
Thank you for your considerations. The power was calculated for a paired T test comparison of a maximal TAT value usually obtained at the end of hemodialysis (i.e. after dialysis from arterial sample) with the web site http://powerandsamplesize.com/Calculators/Compare-2-Means/2-Sample-Equality. The 22 ng/mL were taken from the study by Richtrova (ref. 14), where 2.4 mmol/l of blood citrate was used (36 mmol citrate/h with 250 ml/min blood flow). Our values after 4 h were 8-12 ng/ml, which is in the comparable range. It should be acknowledged that finally, we did not perform several separate paired T tests to compare the 3 time points between the groups, but decided to use repeated measures ANOVA, which tests two hypothesis simultaneously: the overall difference between the groups (our main outcome), and overall increase in time (which is of interest, as if there is no increase, this means we have achieved perfect biocompatibility). For reviewer reference, we provide the paired T test for TAT after 4 hours, arterial sample, corrected for hematocrit, which was 7.5 +/- 6.9 vs. 11.8 +/- 19.9 (standard vs increase dose), with p = 0.47 (please note that values on the graph are least square estimates with SEM, as per ANOVA convention, and not means +/- SD). If the reviewer would insist, we can also report this as the "official" main outcome, but we believe this would not add any value to the study results already presented. Please also mind that other biocompatibility studies were of similar size (e.g. studies ref. 9, 11, 14 had 8-10 patients/procedures included per group, although in cross-over design). We have not decided for crossover design due to considerable volume of sampled blood per patient of prox. 20-30 ml.
Regarding other comments: there was significant difference in duration of RRT treatment (i.e. dialysis vintage, the time from starting hemodialysis to inclusion into the study) and marginally significant difference in age of patients, but this does not affect biocompatibility of an individual hemodialysis procedure. It is possible that this could influence the inflammation burden of patients, affecting values of measured parameters before dialysis. This effect was intended to be filtered out by exclusion criteria (no signs of infection, malignancy, etc). Other than (marginally) non-significant difference in MPO, patients were comparable, and the observed minor difference in MPO is not related to age (i.e. increased dose group is older, but MPO is higher in the standard dose group). The repeated measures ANOVA analysis accounts also for the initial (pre-dialysis) values, when comparing both groups. Regarding the different membrane materials, indeed, there can be minor differences in some aspects of biocompatibility, but synthetic membranes (including polysulfone (FX Cordiax) and polyamix (Polyflux H)) are considered highly biocompatible (as compared to the old cellulose based membranes). This is the reason we have decided to leave each patient included in the study on the dialyzer that he uses for his regular dialysis, as the focus of the study was the effect of anticoagulation (and its dose) on biocompatibility, and not an analysis of a precise membrane type. This would also increase the generalizability of the study, if it would be positive.
As already alluded to by the reviewer, we also do not believe a multivariate analysis would be credible with such a sample size. As the focus of the study was the effect of the dose of anticoagulant (citrate), other differences were left to be annulled by the randomization. Furthermore, the effect of anticoagulation was believed to be far outweighed by other factors. The data obtained (parallel changes in both groups, which are only minor (in absolute values, compared to other studies)) indicate good anticoagulation/biocompatibility in both groups, which was also our conclusion.
"The reviewer understood that Figure 2 is a key result, but it is an insufficient explanation. Figure 2C shows that TAT value at 240 min was greater in increased dose group than in standard dose group. Non-significance between two groups may be caused by small sample size. In addition, the trend of MPO level was similar between two groups, but baseline MPO levels between two groups are observed. The author may point out these results."
Indeed this is the main result and we thank you for noting the small differences between the groups. As explained in the previous point, the difference in TAT at 4 h, arterial sample, was not significant even when directly compared with paired T test (for which an adequate power calculation was made, see previous point) and this is the reason we have not discussed this in the paper. Similarly for the MPO, none of the time points were significantly different (by paired T test) and changes were parallel in both groups. Should there be a group with e.g. a higher release of MPO (e.g. mild infection, with activated leukocytes), the ANOVA test, which tests for all time points simultaneously, would have detected a between group differences. As the main end point was the difference between the groups *during dialysis* (the effect of anticoagulation), and this was not present, minor difference in the initial values was not considered important for discussion.
"Minor comments
Page 2, line 29. TAT may cite the abbreviation."
Thank you, we have corrected that.
"Table 2. Although the author presented accurate biocompatible markers, classical indicators, such as hs-CRP or IL-6s also need to be present."
Unfortunately, those were not taken, so we cannot provide the data. CRP does not change so fast as to increase within 4 hours as a result of bio-incompatibility of a single dialysis procedure, therefore it was not chosen as a marker in the study. IL-6 has a fast response so it could have increased within 4 h of dialysis, but it is not often measured in the before/after dialysis fashion (as it can be removed or adsorbed by certain membranes), but rather in a long-term, as a marker of chronic inflammation. Therefore, we have not chosen it as one of the biocompatibility markers in our "one hemodialysis per patient" study.
"Type of vascular access can influence biocompatibility."
This is debatable, as often patients on catheters are much sicker than patients with an AV fistula, so it is hard to make adequate comparisons. Again, this was not considered a major factor and randomization was performed.
"Page 7, line 4. The author presented that 25% of all showed clean membranes. The reviewer understood that 5 membranes from each group (total 10 membrane) were evaluated in method section."
There is a misunderstanding here. Five dialyzers from both groups (i.e. 10 dialyzers) were prepared for SEM, and 10 randomly selected membrane fibers were taken from each of the ten dialyzers and prepared for SEM, resulting in 100 fiber areas or "samples" for evaluation. 25% of those 100 were completely clean (SEM total score of 0) and this is reported to signify an overall excellent biocompatibility.
"Figure 3. Further explanations for indicators are needed."
We apologize, there was an error in Figure 3 legend. We have added the full text to the revised manuscript. "Figure 3. Scanning electron microscopy of the inner surface of dialyzer fibers with magnified details. A) Dialyzer's inner surface clear of biologic depositions (SEM clotting score of 0), with signs of mechanical damage, most likely caused by tweezers during fiber manipulation (black arrow in A1, A2), and inorganic particles (i.e. debris of fiber wall), most likely produced during cutting (white arrow in A2). B) Minimal depositions of platelets (white arrows in B1, B2) and fibrin ((black arrows in B1, B2) SEM clotting score of 3)."

Reviewer 2 Report
The authors showed the results of an RCT in which they tested two different groups in regional citrate anticoagulation in hemodialysis patients: 1) standard dose of citrate (2.7 mmol / L); 2) higher dose (4 mmol / L). The results and methods are well described, the text is flowing, and the study's characteristics correspond to those described in the database ACTRN12613001340729 (even if the subjects recruited are 32 and not 30 as described in the text). is not enough for publication, I ask Major revisions:
1) this work lacks novelty. The discussion states: "a significant increase in plasma concentration of PF4 was reported during dialysis with 2.4 mmol / l citrate [14], while there was no significant increase in PF4 at 3.4 mmol / l [9] and no increase in beta-thromboglobulin at 4.7 mmol / l [22]. "There are works that have already tested these citrate assays. Why was it necessary to test a higher dose of citrate if the standard one is considered optimal? How would a high level of citrate affect the biocompatibility of dialyzer fibres, according to your hypothesis? Please discuss it.
2) You have tested two different synthetic dialyzers. How were they distributed between the two groups? Did you notice any differences between those who were treated with FX Cordiax and Polyflux H? Show the data by stratifying by type of dialyzer, please.
3) Is it possible to have a quantitative datum of the platelet/fibrin
adherent value from the analysis with scanning electron microscopy considering that the discussion states: "The SEM clotting score was excellent and showed almost completely clean fibres in both citrate dose groups."? I would like to see images of the fibres of the two groups being treated and of both FX Cordiax and Polyflux H membranes.
4) In the paragraph "Biocompatibility parameters", the authors state: "We did not observe leukopenia or thrombocytopenia during hemodialysis in either group (data not shown)." Could you show these data, considering that in the discussion, the authors state: "during hemodialysis with cellulose membranes, causing leukopenia and dyspnea."?
5) In discussion, it is also stated about this study: "The main limitation is a modest number of included patients and the use of different synthetic membranes." If you decided to do a clinical trial with 30 participants, did you choose from the beginning to do a study with this limit? If the number guaranteed an excellent power of the study, how can this be a statistic limit?
6) In the discussion, it states: "It should be noted that using a very high dose of citrate might increase the possibility of side effects and citrate accumulation, although hemodialysis provides excellent clearance of citrate." Would it be possible to see a citrate assay in the plasma samples to demonstrate no citrate accumulation? Which side effects have you observed?
Author Response
Point by point reply to Reviewers - Reviewer 2
"The authors showed the results of an RCT in which they tested two different groups in regional citrate anticoagulation in hemodialysis patients: 1) standard dose of citrate (2.7 mmol / L); 2) higher dose (4 mmol / L). The results and methods are well described, the text is flowing, and the study's characteristics correspond to those described in the database ACTRN12613001340729 (even if the subjects recruited are 32 and not 30 as described in the text). It is not enough for publication, I ask Major revisions:
1) this work lacks novelty. The discussion states: "a significant increase in plasma concentration of PF4 was reported during dialysis with 2.4 mmol / l citrate [14], while there was no significant increase in PF4 at 3.4 mmol / l [9] and no increase in beta-thromboglobulin at 4.7 mmol / l [22]." There are works that have already tested these citrate assays. Why was it necessary to test a higher dose of citrate if the standard one is considered optimal? How would a high level of citrate affect the biocompatibility of dialyzer fibers, according to your hypothesis? Please discuss it."
Thank you for your comments. There are indeed several studies on the biocompatibility of regional citrate anticoagulation during hemodialysis in the literature, but they are usually comparing citrate with heparin anticoagulation. Furthermore, the dose of citrate used differs by studies and dialysis centers, as discussed in the introduction. From the analysis of published data, it could be assumed (and this is also logical from the mechanistic point), that higher doses of citrate would confer better biocompatibility, but (to our knowledge) there is no direct comparison of two citrate doses in the literature. Comparison of different doses between different studies is difficult, since there could be other differences, e.g. variance in the technical/practical aspects of hemodialysis (e.g. flushing the circuit with heparin, preparation of the extracorporeal circuit, pump speed (which can induce generation of bubbles), and many little details, which can influence biocompatibility and activation of hemostasis) and also in the laboratory methods. Therefore, this is the first study to directly compare the dose of citrate that was considered "standard" (i.e. the one most often reported in the literature and also the one used in routine clinical practice in our center) to a "higher" dose, for which there were some data in the literature, that it could confer even better biocompatibility, due to more sever hypocalcemia in the extracorporeal circuit. The standard dose is considered very efficient in the everyday clinical practice, but some clinically undetectable bio-incompatibility could be present and could in theory have adverse effects on the inflammatory and atherosclerotic processes in the patient, as discussed. It is therefore important to find an optimal dose of citrate to be used in everyday clinical practice and it is a novel observation that the dose the majority of centers are using seems sufficient also when the biocompatibility is comprehensively evaluated.
The higher dose of citrate affects the biocompatibility of the membrane by inducing more severe hypocalcemia, which reduces the blood-membrane interactions (citrate is also used in the blood collection tubes to prevent activation of blood cells after sampling) as has been repeatedly shown in in-vitro studies on citrate anticoagulation and can also be conferred from comparing studies using different doses of citrate as discussed in the discussion and cited by the reviewer. Therefore, citrate anticoagulation is able to overcome the residual bio-incompatibility of the membrane material itself. More explicit statement about this is added to introduction, while the differences for specific biocompatibility markers are already discussed in the discussion. Our prior in-vitro work (ref. 17) has indeed shown some improvements with higher dose, and it was our intention to test this also in a clinical study.
"2) You have tested two different synthetic dialyzers. How were they distributed between the two groups? Did you notice any differences between those who were treated with FX Cordiax and Polyflux H? Show the data by stratifying by type of dialyzer, please."
Indeed, there are many aspects of hemodialysis procedure that influence its biocompatibility and membrane material is one of them. Synthetic membranes (including polysulfone (FX Cordiax) and polyamix (Polyflux H)) are considered highly biocompatible (as compared to the old cellulose based membranes) with only minor differences between the materials. This is the reason we have decided to leave each patient included in the study on the dialyzer that he uses for his regular dialysis. As the focus of the study was the effect of anticoagulation (and its dose) on biocompatibility, we let the other factors influencing variate and let them be neutralized by the randomization. Given the small sample size in the study, we believe stratification is not possible. Furthermore, since all the observed changes in biocompatibility parameters were parallel in both groups, this suggests that there is some common factor between the groups affecting this and not an effect of a subgroup (of two different dialyzers), as this would require this factor to be very strong, to show its effect in approximately half the sample size, which is already somewhat small.
"3) Is it possible to have a quantitative datum of the platelet/fibrin adherent value from the analysis with scanning electron microscopy considering that the discussion states: "The SEM clotting score was excellent and showed almost completely clean fibers in both citrate dose groups."? I would like to see images of the fibres of the two groups being treated and of both FX Cordiax and Polyflux H membranes."
As explained also to the other reviewer, there were five dialyzers from both groups taken and 10 randomly selected fibers from each dialyzer prepared for SEM resulting in 100 fiber areas or "samples" for evaluation by SEM. Each of the 100 fiber areas was evaluated with a semi-quantitative total clotting score (as explained in the methods) and the data are reported as median and inter-quartile range for each of the two groups and there was no between-group difference. To explain the absence of a difference, it is noted that 25% of those 100 were completely clean and this is reported to signify an overall excellent biocompatibility. The median SEM score was 2 in both groups and the theoretical worst score would be 20 points, so the fiber were very "clean". This is the reason, why the overall interpretation in the discussion and abstract was that "The SEM clotting score was excellent and showed almost completely clean fibers in both citrate dose groups".
These numerical results of 100 fiber samples are illustrated with two representative SEM images, to illustrate what was done and to make the SEM scores more visually comprehensible. One figure is showing a completely clean fiber (score of 0 points, panel A) and a "typical" fiber with a score of 3 points (the most typical score was 2 or 3 points). As a median SEM score was identical in both groups, no "by group" comparison of image is possible and two representative values of the SEM score are illustrated. Comparison by the membrane type was not an outcome of the study, which focused on the dose of anticoagulant, meant to overcome any remaining incompatibility of the membrane (and the extracorporeal system as a whole).
Of note, there was one technical error and part of the Figure 3 legend was missing and is now replaced.
Access to all 100 SEM images is possible, if the reviewer requests this, they can be uploaded to Google drive and shared with reviewer, if an email of the reviewer is provided.
"4) In the paragraph "Biocompatibility parameters", the authors state: "We did not observe leukopenia or thrombocytopenia during hemodialysis in either group (data not shown)." Could you show these data, considering that in the discussion, the authors state: "during hemodialysis with cellulose membranes, causing leukopenia and dyspnea."?"
We apologize for our omission of a formal statistical analysis of leukocyte and thrombocyte data. We have now performed a repeated measures ANOVA analysis, as for other parameters, and added the data to Figure 2 as requested. There is indeed a mild reduction in leukocytes, which is a well known phenomenon and is considered clinically insignificant (leukopenias described with cellulose membranes were to a significant level). There are no changes in thrombocytes, taken from the arterial blood, and a small increase in thrombocytes over the dialyzer (from arterial to venous sampling site at 4 h of dialysis), for which we have no explanation (if anything, there should be a reduction of thrombocytes, as a result of membrane adhesion) and therefore believe it to be a chance finding.
"5) In discussion, it is also stated about this study: "The main limitation is a modest number of included patients and the use of different synthetic membranes." If you decided to do a clinical trial with 30 participants, did you choose from the beginning to do a study with this limit? If the number guaranteed an excellent power of the study, how can this be a statistic limit?"
The decision to include different membranes (the one the patient was using outside of the study) was made for practical reasons, but also to the make the study more generalizable (if we would obtain a positive result in a study with only one dialyzer, it could be argued, that it does not apply to other membranes). Sample size was listed as a study limitation despite the fact that the power was calculated and was appropriate, since the power was calculated (appropriately) according to only one of the measured parameters (TAT at 4 h of dialysis (arterial sample), the one, for which there is most data, and which was considered the most "basic" biocompatibility parameter), but since the sample was relatively small (as noted by both of the reviewers), it is possible, that a larger sample would be needed to detect a difference in one of the other parameters measured. So although the study had adequate power and the decision on membranes was thought of in advance, we still consider this a legitimate critique of the study and have added this to limitations.
"6) In the discussion, it states: "It should be noted that using a very high dose of citrate might increase the possibility of side effects and citrate accumulation, although hemodialysis provides excellent clearance of citrate." Would it be possible to see a citrate assay in the plasma samples to demonstrate no citrate accumulation? Which side effects have you observed?"
We did not measure citrate concentrations in the blood, so we cannot provide data on that. Concentrations during dialysis are usually in the 0.5 - 1 mmol/l range (2-10x above normal values). Also, it is not clear, what is the cut-off value for "citrate accumulation/intoxication", as the clinical effects of citrate are mediated by ionized hypocalcemia, which is corrected independently of serum citrate concentration with calcium infusion and iCa measurements. Citrate accumulation is usually indirectly measured by the ratio between total and ionized calcium (which resembles serum citrate concentration, since citrate chelates calcium (doi: 10.1097/00003246-200104000-00010)) and this was not increased. We have added data on total-to-ionized calcium ratio after dialysis and % patients with increased value to the manuscript (Table 3).
As written in the results, there were some cases of mild metabolic alkalosis (which can be attributed to citrate being metabolized to bicarbonate) and one episode of hypocalcemia, which is the major (and expected) complication when using citrate; this is also the reason why ionized calcium needs to be monitored during citrate dialysis.

Round 2
Reviewer 1 Report
I have no additional comment.